

# A dataset of 190 lakes' ice phenology on the Tibetan Plateau extracted from AMSR2 data

Yu Cai[1], Jingjing Wang[1], Yao Xiao[1], Zifei Wang[1,2], Xiaoyi Shen[1], Haili Li[1], Chang-Qing Ke[1]

[1]Jiangsu Provincial Key Laboratory of Geographic Information Science and Technology, Key Laboratory for Land Satellite Remote Sensing Applications of Ministry of Natural Resources, School of Geography and Ocean Science, Nanjing University, Nanjing 210023, China
[2]Aerospace Times Feihong Technology Company Ltd., Beijing 100000, China

*Correspondence to*: Chang-Qing Ke (kecq@nju.edu.cn)

**Abstract.** The utilization of passive microwave land–water mixed pixels in extracting lake ice phenology has been underestimated. There are still many small and medium-sized lakes whose ice phenology has not been recorded, especially in regions such as the Tibetan Plateau, for which there are very few in situ observations. In this study, the changing characteristics of mixed pixels during freeze–thaw processes are discussed. Using air temperature to reduce the seasonal variation in brightness temperature series, Advanced Microwave Scanning Radiometer 2 (AMSR2) data were used to extract the ice phenology of 194 lakes that could represent 78% of the total lake area on the Tibetan Plateau from September 2012 to August 2022. The lake ice phenology results show high consistency compared with the Moderate Resolution Imaging Spectroradiometer (MODIS)-derived lake ice phenology dataset (the average MAEs range from 4.5 to 8.4 days for different freeze–thaw events) and demonstrate better performance than the existing AMSR2-derived dataset. This ice phenology dataset covers the largest number of lakes in the Tibetan Plateau region, fills a gap or increases the integrity of ice phenology records for at least one hundred small lakes and provides complete pixel-scale freeze–thaw processes on the lake surface. The dataset will provide valuable information to the user community about the spatial distribution of and changes in ice cover in lakes, especially small lakes, over the last decade. The dataset is available at https://doi.org/10.11888/Cryos.tpdc.300796 (Cai and Ke, 2023).

## 1 Introduction

While there are numerous lakes with regular ice cover on the Tibetan Plateau, historical freeze-up and break-up records are available for few of them due to harsh natural conditions. Meanwhile, there are few meteorological stations on the Tibetan Plateau near lakes. In response to weather and climate conditions, lake ice can serve as a good indicator of local weather and regional climate (Maslanik and Barry, 1987; Magnuson et al., 2000; Benson et al., 2012; Dörnhöfer and Oppelt, 2016; Sharma et al., 2019). Therefore, many studies have used remote sensing data to obtain long time series and large spatial scale lake ice records (e.g., Kropáček et al. 2013; Wang et al., 2021a; Wu et al., 2021; Zhang et al., 2021a; Su et al., 2021; Cai et al., 2022a).



Various remote sensing data capable of providing daily observations have been used to extract lake ice phenology on the Tibetan Plateau, including optical data, such as Advanced Very High Resolution Radiometer (AVHRR, Latifovic and Pouliot, 2007; Weber et al., 2016) data, Moderate Resolution Imaging Spectroradiometer (MODIS, Nonaka et al., 2007; Kropáček et al., 2013; Cai et al., 2019; Wu et al., 2021) data, and passive microwave data, such as Scanning Multichannel Microwave

Radiometer (SMMR) data, Special Sensor Microwave Image (SSM/I) and Special Sensor Microwave Imager/Sounder (SSMIS) data (Ke et al., 2013; Cai et al., 2017; Su et al., 2021; Cai et al., 2022a), Advanced Microwave Scanning Radiometer for Earth Observing System (AMSR-E) and Advanced Microwave Scanning Radiometer 2 (AMSR2) data (Kang et al., 2012; Du et al., 2017), and Microwave Radiation Imager (MWRI, Wang et al., 2021a) data. We summarize all of the available lake ice phenology on the Tibetan Plateau or on the global scale (Table 1).


**Table 1 Summary of literature examining lake ice phenology on the Tibetan Plateau.**

| Reference | Sensor | Study area & Number of lakes on the Tibetan Plateau | Method | Object |
|---|---|---|---|---|
| Kropáček et al., 2013 | MODIS | Tibetan Plateau, 59 lakes | Threshold | Selected lakes |
| Cai et al., 2022b | MODIS | Tibetan Plateau, 71 lakes | Threshold | Selected lakes |
| Du et al., 2017 | AMSR-E, AMSR2 | Global, 13 lakes* | Threshold | Pixels with water coverage $\geq 90\%$ |
| Cai et al., 2022a | SMMR, SSM/I, SSMIS | Global, 8 lakes | Threshold | Pixels 6.25 km away from lake shore |
| Su et al., 2021 | SMMR, SSM/I, SSMIS | Global, 3 lakes | Visual interpretation | Pixel of the center point of lakes $\geq 625$ km$^2$ |
| Wang et al., 2021 | SMMR, SSM/I, AMSR-E, MWRI, AMSR2 | Global, 106 lakes | Visual interpretation | Pixel of the center point of lakes $\geq 40$ km$^2$ |

*Du et al. (2017) did not provide lake-scale records; the number of lakes was estimated by authors from classified water/ice pixels.

For medium- and high-resolution optical data, after classifying water and ice pixels, the dates of freeze-up and break-up can be extracted by setting thresholds (Kropáček et al., 2013; Yao et al., 2016; Cai et al., 2019). Mathematical models have also

been used to simulate changes in the number of water/ice pixels during the freeze–thaw process to estimate lake ice phenology (Latifovic and Pouliot, 2007; Zhang et al., 2021b). In addition to using only reflectance products, lake ice phenology can also be extracted by combining LST products (Weber et al., 2016; Guo et al., 2018). However, no matter what product or method



has been used, lake ice phenology extracted from optical data has always presented uncertainties due to the presence of clouds (Cai et al., 2019; Murfitt and Duguay, 2021).

On the other hand, while passive microwaves have the advantage of not being affected by weather and illumination conditions, their application has been greatly limited by their coarse spatial resolution (several km to tens of km, Duguay et al., 2015). Generally, accurately extracting lake ice phenology from the brightness temperature ($T_B$) series of a pixel using threshold-based algorithms requires a large enough difference in the $T_B$ between the ice-free period and ice-covered period; that is, the pixel should be sufficiently pure. For example, when Du et al. (2017) extracted global lake ice phenology from

AMSR-E and AMSR2 data, only pixels with ≥ 90% water coverage were screened. Similarly, Cai et al. (2022a) eliminated SMMR and SSM/I-SSMIS pixels within 6.25 km of a lake boundary to reduce the impact of land contamination. Wang et al. (2021a) only used a pixel closest to the center of a lake to extract ice phenology from multiple passive microwave sensors, including SMMR, SSM/I, AMSR-E, MWRI, and AMSR2 sensors. As a result, passive microwaves have been used to obtain lake ice phenology only for a limited number of large (area > 500 km$^2$) or medium (50–500 km$^2$) lakes. In fact, for large

quantities of pixels located on the shores of lakes or covering smaller lakes (< 50 km$^2$), their $T_B$ change patterns also include freeze–thaw information.

  To date, there has been no passive-microwave-derived lake ice phenology dataset that focuses on small lakes, and there has been little discussion about the $T_B$ characteristics present during freeze–thaw processes of land–water mixed pixels. Therefore, this study will re-explore the application potential of passive microwave data in the field of lake ice phenology. Taking the

Tibetan Plateau, where small and medium-sized lakes are concentrated, as an example, AMSR2 $T_B$ and ERA5 air temperature data are used to extract lake ice phenology on the Tibetan Plateau from September 2012 to August 2022.

## 2 Data and methods

### 2.1 Study lakes

Lakes on the Tibetan Plateau have been expanding in recent decades (Yao et al., 2018; Zhang et al., 2021c). Therefore, to

extract the ice phenology of as many lakes on the Tibetan Plateau as possible, the latest lake boundary – the boundary of lakes in 2022 (Zhang et al., 2019) – was used. The dataset contains boundary for lakes larger than 1 km$^2$ from 1970s to 2022 delineated from Landsat data by using Normalized Difference Water Index (NDWI). For each lake larger than 10 km$^2$, we drew a 20 km buffer outward from the boundary and determined whether the AMSR2 pixels within the buffer contained extractable freeze–thaw information. A relatively large buffer was used because the freeze–thaw information of a lake might

be contained in adjacent pixels since the original resolution of the AMSR2 footprint is 7 × 12 km, while on the other hand, the extension of a lake might change during this decade. Furthermore, although altitude correction had been processed in the AMSR2 Level-1R (L1R) products, slight terrain errors could still be observed in the Tibetan Plateau region. Only pixels that provided complete and valid freeze–thaw information from September 2012 to August 2022 were retained, and as a result, the ice phenology of a total of 194 lakes was extracted in this decade (Fig. 1). These lakes accounted for ~ 45% of all lakes > 10



km² on the Tibetan Plateau, and their area accounted for ~ 78% of the total lake area on the Tibetan Plateau. Lakes not included might be too small or narrow, have complex shorelines, or do not form annual ice cover due to low latitude or high salinity. The earliest freeze-up of the lakes on the Tibetan Plateau usually begins in October or November, and the latest break-up occurs in June or July, so the lake ice period is determined to run from September to August of the following year.

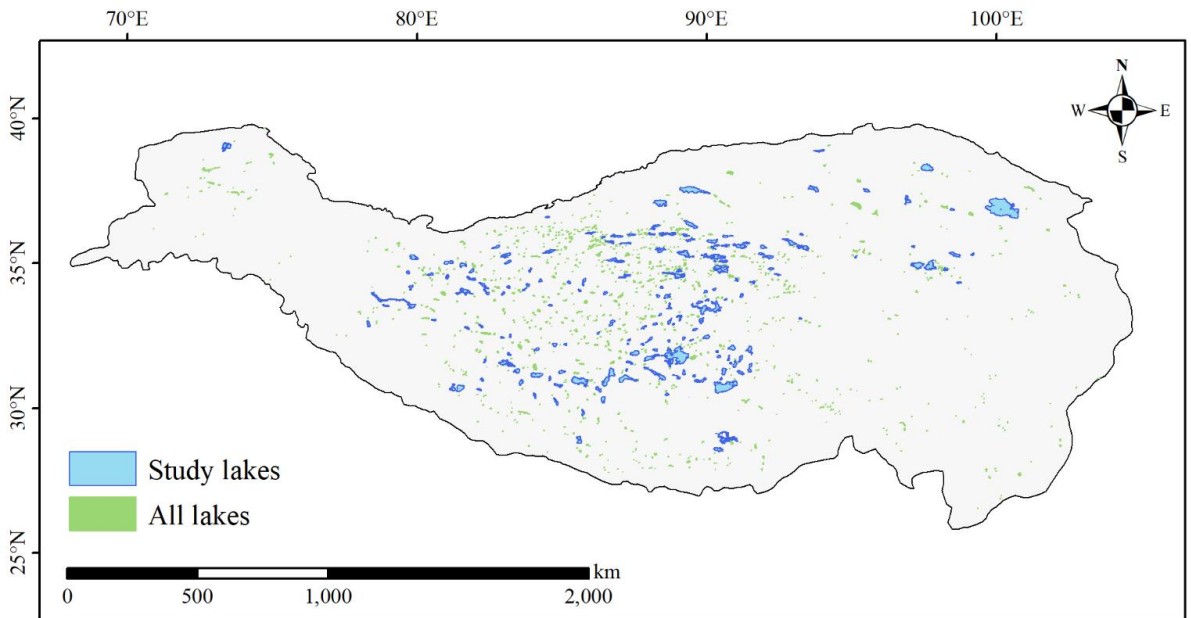

**Figure 1: Locations of the 194 study lakes and all lakes larger than 1 km² on the Tibetan Plateau.**

## 2.2 Input data

### 2.2.1 AMSR2 L1R product

AMSR2 L1R swath brightness temperature ($T_B$) data (36.5 GHz horizontal polarization ascending data with a spatial resolution

of 10 km) for September 2012 to August 2022 were obtained from Japan Aerospace Exploration Agency (JAXA, 2012). The $T_B$ data in the L1R product were spatially resampled from the L1B product to match the center position and size of the field of view of each frequency band at each pixel, which is very important for high-altitude areas, including the Tibetan Plateau, because a high altitude can cause displacements of the geographical location of AMSR2 pixels (Maede et al., 2016). In the latitudinal zone of the Tibetan Plateau, multiple swath 36.5 GHz $T_B$ data for the same day do not overlap; therefore, we simply

collected all L1R data covering the Tibetan Plateau to obtain the 10-year $T_B$ sequence for each lake pixel.





### 2.2.2 ERA5 air temperature

The ERA5 2 m temperature was also used to extract lake ice phenology. The spatial resolution is 0.1°, precisely matching the AMSR2 pixel. We collected hourly data (Muñoz Sabater, 2019) and calculated daily averages to obtain the 10-year air temperature series for each lake pixel.

### 2.2.3 Lake ice phenology datasets

Two lake ice phenology datasets – one extracted from the MODIS daily snow product (denoted as MODIS LIP, Cai and Ke, 2022) and the other extracted from multiple passive microwave data (denoted as PMW LIP, Wang et al., 2021b) – were used to compare them to the lake ice phenology results extracted from AMSR2 data from September 2012 to August 2020. These two datasets contained the most lakes among the existing lake ice phenology datasets extracted using optical data and passive microwave data (Table 1). Both datasets provided records of four freeze–thaw events: the start and end of freeze-up and the start and end of break-up. The MODIS LIP provided ice phenology for 71 lakes on the Tibetan Plateau for 2001 to 2020, with the ice phenology of 64 of these lakes also extracted by AMSR2 data. MODIS showed greater advantages in extracting the ice phenology of lakes with smaller areas, but its accuracy might suffer from cloud cover (Cai et al., 2019).

PMW LIP was produced from multiple passive microwave data, including SMMR, SSM/I, AMSR-E, MWRI, and AMSR2 data. Records that overlap with the temporal coverage of this study (September 2012 to September 2020) were also from AMSR2 L1R data, but 18.7 GHz data with a spatial resolution of $22 \times 14$ km were used. Only the passive microwave pixel closest to the central point of a lake was used to extract ice phenology to represent an entire lake (Wang et al., 2021a). The dataset covered 106 lakes on the Tibetan Plateau, 105 of which were included in this study. Due to the limited ability of the original $T_B$ data to reflect freeze–thaw information, especially for small lakes covered by mixed passive microwave pixels, not all of the ice phenology records of these 106 lakes were complete.

### 2.3 Methods

### 2.3.1 $T_B$ characteristics of different types of pixels

Pure land pixels and pure water pixels with freeze–thaw processes showed great differences in their $T_B$ variations over a year (Fig. 2). For pure land pixels, the change pattern of $T_B$ is quite similar to that of air temperature, with high temperatures in the summer and low temperatures in the winter (Fig. 2b). However, for pure water pixels, the low emissivity of water results in a very low $T_B$ in the ice-free season, while the high emissivity of ice leads to a high $T_B$ in the ice-covered season (Fig. 2d, Kang et al., 2012). Due to the rapid $T_B$ changes caused by freeze-up and break-up processes, threshold-based or mutation-detected methods have been successfully performed to extract the freeze–thaw events of pure water pixels (Ke et al., 2013; Cai et al., 2017; Du et al., 2017). However, for land–water mixed pixels that are widespread on the shores of lakes or that cover small lakes, there is a lack of research on $T_B$ change characteristics and freeze–thaw event extraction methods. Land–water mixed pixels have the characteristics of both pure land pixels and pure water pixels; that is, the $T_B$ will fluctuate seasonally, and the

presence of ice will increase the $T_B$. Since the $T_B$ continuously fluctuates during the ice-free season and increases in the summer as the air temperature rises, the ice-covered season does not necessarily present a higher $T_B$, especially for pixels with a high proportion of land (Fig. 2c). Although freeze–thaw information is clearly included in the $T_B$ changes, under such circumstances,

it is difficult to find a threshold to classify ice and water based on the original $T_B$ series.

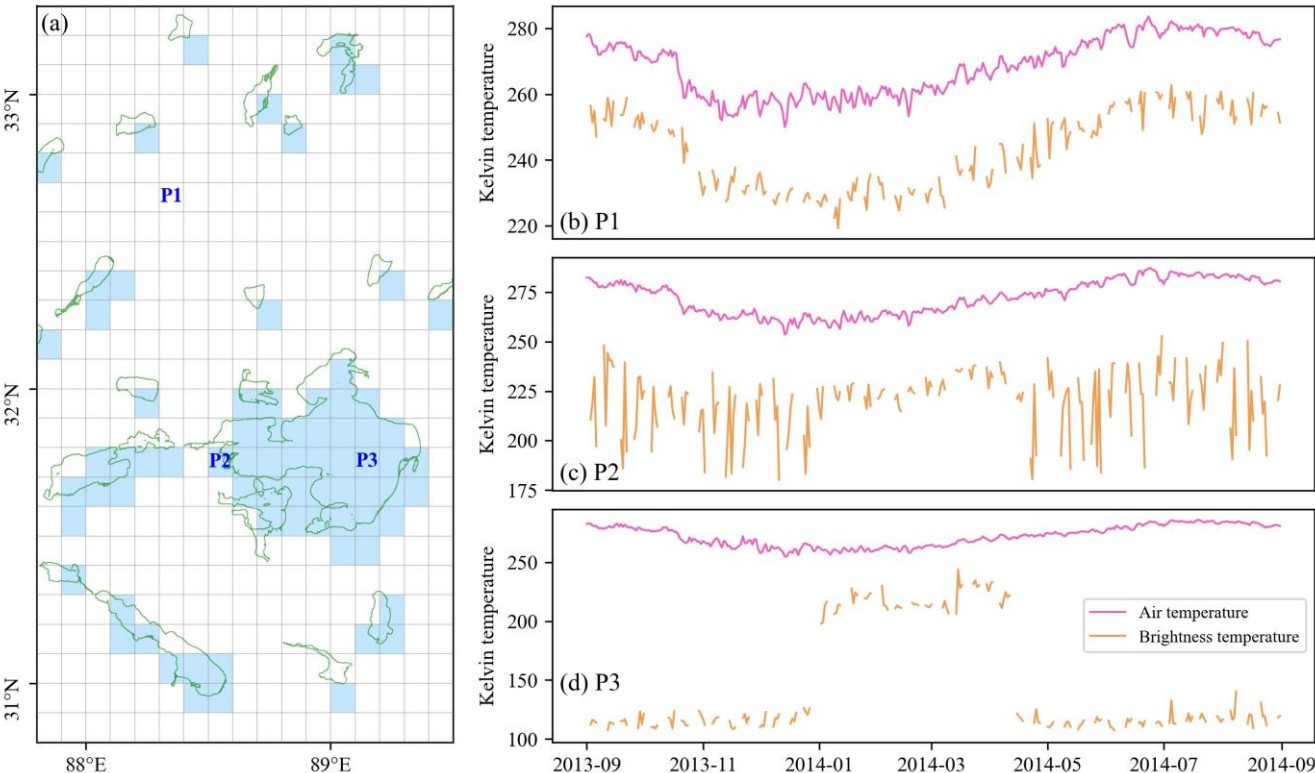

**Figure 2: Samples of pure land pixels (P1), mixed pixels (P2), and pure water pixels (P3). (a) Locations of selected sample pixels, (b), (c), and (d) air temperature and brightness temperature series from September 2013 to August 2014 for the P1, the P2, and the P3, respectively. Slight displacements in the geographical positions of the pixels were caused by terrain error.**


Visual interpretation can always be used to extract the freeze–thaw events of these pixels, but this approach is inevitably time-consuming and subjective. Therefore, to increase the consistency of the records between years and pixels, we performed some simple processing on the original $T_B$ series to make the threshold method feasible again. Manual visual interpretation will be used afterward to check and modify the results of the threshold method and increase the integrity of the lake ice

phenology records.

Earth System
Science
Data

### 2.3.2 Extraction of lake ice phenology

The extraction of lake ice phenology is performed independently for each pixel. To set a threshold to divide ice and water, the $T_B$ during the ice-covered period is expected to be higher than that during the ice-free period. However, mixed pixels with a high proportion of land may have a higher $T_B$ in the summer than in the winter. Therefore, first, air temperature data were used to reduce the seasonal variation in the $T_B$ series. We calculated a ratio of $T_B$ to air temperature in ice-free season by dividing the average $T_B$ by the average air temperature in September, July, and August (for example, for the period of 2013/2014, this refers to September 2013 and July and August 2014). The higher the proportion of water in the pixel is, the lower the ratio is. Meanwhile, the air temperature series was fitted by a cubic polynomial function. Then, the air temperature fitting line was multiplied by the ratio to obtain the "fitting line" of the $T_B$ series (Fig. 3a). Afterward, a seasonal-variation-reduced $\Delta T_B$ series could be calculated by subtracting the fitting $T_B$ series from the original $T_B$ series. During ice-free periods, the $\Delta T_B$ fluctuates by approximately 0, while during ice-covered periods, the $\Delta T_B$ increases substantially (Fig. 3b).

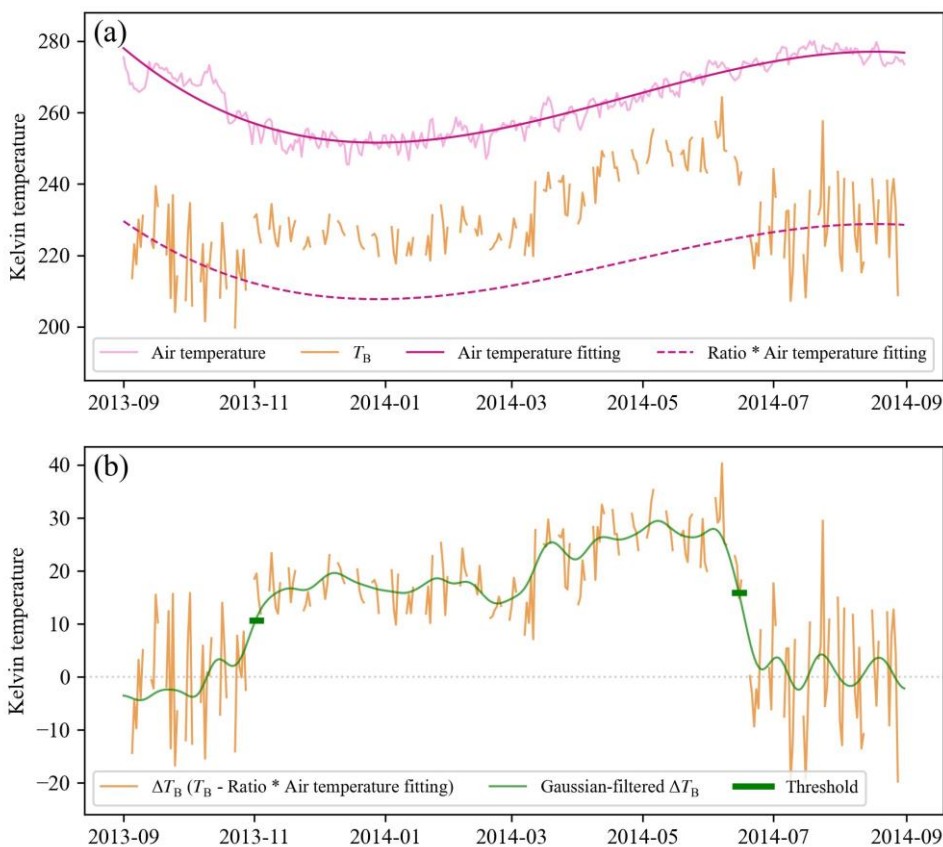

**Figure 3: Example of the conversions of air temperature and $T_B$ series and the calculation of thresholds for a pixel of Group Hoh Xil (91.4°E, 35.7°N) for the 2013/2014 period.**




The $T_B$ and hence the $\Delta T_B$ of the mixed pixels fluctuate greatly in the time series, especially during the ice-free period; in addition, there are periodic missing values of $T_B$ observations on the Tibetan Plateau. Therefore, we used linear interpolation to fill in the missing values of the $\Delta T_B$ series and a one-dimensional Gaussian filter to obtain a smoother series. As shown in Fig. 3b, the Gaussian-filtered $\Delta T_B$ showed a clear difference between ice-free and ice-covered periods. Subsequently, following

the method for calculating the reference values used in Cai et al. (2019), we obtained the maximum and minimum values throughout the year and calculated their mean values to divide the Gaussian-filtered $\Delta T_B$ series into two groups. Then, the mean values of the two groups were calculated. Finally, the threshold (TH) was calculated by averaging the two mean values.

This TH performed well in extracting freeze-up dates, but it was not perfect in the case of break-up dates. As ice formation requires colder temperatures than decay, temperatures (both air temperature and $T_B$) during break-up periods tend to be higher

than those during freeze-up periods. In addition, as ice thickness increases in the winter, the $T_B$ will increase further. As a result, even if seasonal variation were reduced for the $\Delta T_B$ series, the overall series might still be tilted, especially for pixels with longer ice periods (Fig. 3). Therefore, the threshold used to determine ice status during the break-up periods should be slightly higher than the threshold used for freeze-up periods. Taking the 2013/2014 period as an example, we manually extracted the freeze-up and break-up dates for 904 pixels (the number of pixels and freeze-up/break-up results might differ

from the dataset used in this study because different extraction criteria were used) and found that the Gaussian-filtered $\Delta T_B$ of the dates of freeze-up and break-up had a significant linear relationship with the ratio, indicating that for pixels with a higher proportion of water, the threshold of Gaussian-filtered $\Delta T_B$ would decrease accordingly. In addition, the thresholds for break-up periods were always higher than those for freeze-up periods (Fig. 4). Therefore, based on the two linear fitting lines, we estimated a correction value of $30 - ratio \times 30$ to be added to the TH as the threshold for break-up periods (TH_corr).





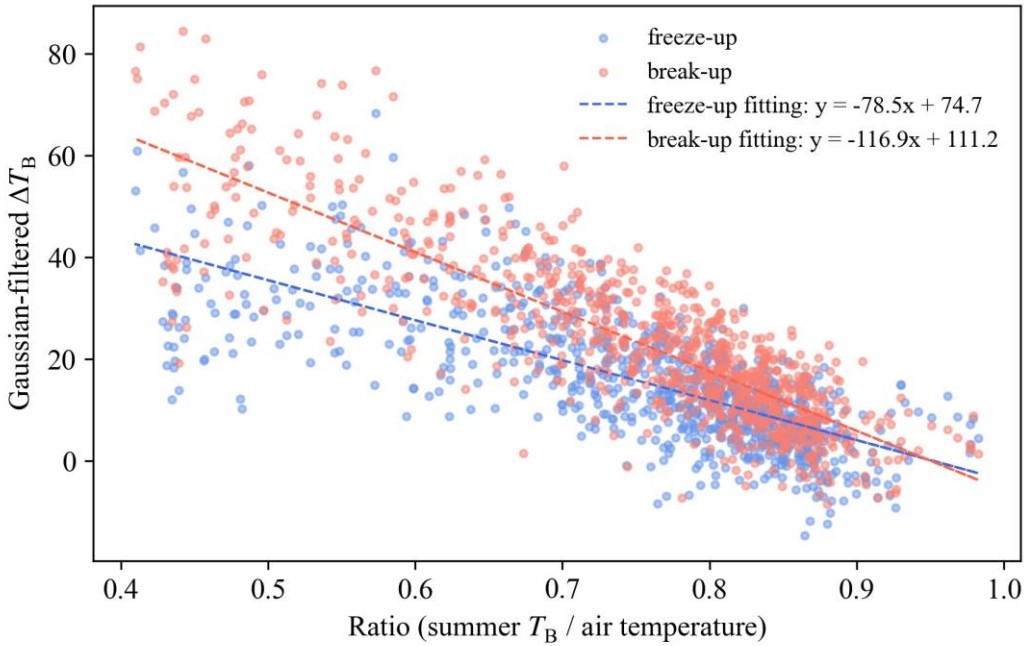


**Figure 4: The scatter and linear fitting line between the ratio and Gaussian-filtered $\Delta T_B$ of manually extracted freeze-up and break-up dates. The Gaussian-filtered $\Delta T_B$ of the date of break-up was always higher than that of freeze-up.**

### 2.3.3 Modification of results and establishment of the dataset

The first day on which the Gaussian-filtered $\Delta T_B$ was greater than the TH during the freeze-up period was recorded as the freeze-up date. If the original $T_B$ of that date was missing, the first date after that day with a valid value was recorded. For the break-up date, the first day on which the Gaussian-filtered $\Delta T_B$ was lower than TH_corr was recorded, and the latest date before that day with a valid value was recorded if the original $T_B$ was missing. Afterward, as mentioned above, all of the freeze-up and break-up dates were checked and modified by manual visual interpretation, and only pixels with complete 10-year

freeze-up and break-up records were retained unless the entire lake in which a pixel was located did not form ice cover in certain years. The integrity of the pixel records needed to be ensured because initial ice formation and final decay often occur near lake shores, where highly land-contaminated mixed pixels are present. Ice in these pixels sometimes persisted for a longer period than other pixels in the given lake, but freeze–thaw events might be difficult to determine for some years. In this case, the presence/absence of records from these pixels would have a great impact on the overall freeze-up/break-up results of the

given lake, resulting in less comparability of results between years.

Finally, 749 pixels were retained and carefully matched to the lake boundary. The names of the lakes were obtained from the HydroLAKES dataset (https://www.hydrosheds.org/page/hydrolakes, last access: 3 December 2019). For unnamed lakes, the longitude and latitude of one of the pixels of the given lake were marked instead (for example, a lake named long888lat333

denotes that it contains a pixel at 88.8°E, 33.3°N). Due to the relatively coarse spatial resolution of AMSR2 pixels, sometimes
a pixel is located between two (or more) lakes. The pixel can contain freeze–thaw information for both lakes, making it difficult
to determine which lake the pixel belongs to. In this case, these lakes are divided into a group, and the name of the lake group
is marked by adding a "Group" before the name of one of the lakes in the group. As a result, the 749 pixels were matched to
153 lakes/lake groups, including 121 individual lakes and 32 lake groups, for a total of 194 lakes.

### 2.3.4 Comparison with existing lake ice phenology datasets

We compared two datasets based on passive microwave data (our dataset, denoted as AMSR2 LIP, and PMW LIP) to the
dataset extracted from optical data (MODIS LIP). Both the PMW LIP and MODIS LIP datasets provided records of four lake
ice phenology events – the beginning and end of freeze-up and break-up, but with different terms (the terms freeze onset,
complete ice cover, melt onset, and complete ice free were used in PMW LIP, and the terms freeze-up start, freeze-up end,
break-up start, and break-up end were used in MODIS LIP). We unified the four events as freeze-up start (FUS), freeze-up
end (FUE), break-up start (BUS), and break-up end (BUE) for our comparison.

The time range of the comparison ran from September 2012 to August 2020. In the PMW LIP dataset, complete freeze–
thaw event records are not available for all lakes. Therefore, to draw the comparison with the MODIS LIP dataset, for each
event, lakes with fewer than six records for the eight-year period were removed. For the AMSR2 LIP, to compare with the
four freeze–thaw events, for each lake (or lake group), the earliest freeze-up date of all pixels was considered the FUS date,
the latest freeze-up date was considered the FUE date, and the earliest and latest break-up dates were considered the BUS and
BUE dates, respectively. Fifty lakes common to the three datasets were screened for comparison. Since the AMSR2 LIP dataset
contains lake groups and some lakes connected into one lake in the lake boundary of 2022 (Zhang et al., 2021c), it is possible
that one AMSR2 lake/lake group corresponds to two or more PMW lakes and MODIS lakes. Specifically, the 50 AMSR2
lakes/lake groups correspond to 65 PMW lakes and 57 MODIS lakes. If one AMSR2 lake/lake group corresponded to multiple
PMW or MODIS lakes, the earliest freeze-up and break-up dates among these lakes would be obtained as the new FUS and
BUS, and the latest freeze-up and break-up dates would be obtained as the new FUE and BUE, respectively. For each of the
50 lakes/lake groups, the correlation coefficient, mean difference, and mean absolute error between the AMSR2 LIP and
MODIS LIP and between the PMW LIP and MODIS LIP were calculated.

## 3 Results and discussion

### 3.1 Uncertainty and cross-validation

There are two main sources of error in the ice phenology extracted from AMSR2 data: the failure to identify the freeze–thaw
process occurring inside the pixels and periodically missed $T_B$ observations. Therefore, freeze-up and break-up dates might
occur later than the actual beginning of the freeze-up and break-up of a pixel but before the actual end date. For mixed pixels
with a high proportion of land, it is difficult to extract more detailed freeze–thaw information. Instead, we focused on the



increase in the number of research lakes and on the comparability of results during the 10-year period. For periodic missing
data, we recorded the earliest freeze-up date and the latest break-up date, which might also result in uncertainty. However, in
the Tibetan Plateau region, even for lakes at the lowest latitude, the sampling interval of the AMSR2 data generally only
spanned one day. That is, the error caused by periodic missing data would not exceed one day.

In addition, when compared to existing lake ice phenology datasets, differences might occur due to the different definitions
of lake ice phenology, different lake boundary datasets used, and possible changes in lake extents in recent decades.

### 3.1.1 Definition of lake ice phenology events

The lake ice phenology in this study was extracted at the pixel scale, and only two events – freeze-up and break-up – were
extracted. Freeze-up represents that the pixel has been covered by stable lake ice for a certain extent, and break-up represents
that certain extent of water can be observed inside the pixel. The definitions of these two events are similar to those of the
terms "ice dominant" and "water dominant" used by Du et al. (2017) to describe the lake ice status determined by a threshold
method based on the moving t test. However, the terms do not have exactly the same meaning as the terms "ice on" and "ice
off", which are usually used in in situ observations to indicate the first day of complete ice cover and the last day before
completely ice-free conditions, respectively (Magnuson et al., 2000). Compared to the four events defined in Cai and Ke (2022)
and Wang et al. (2021b), which emphasize the beginning and end of the freeze-up and break up, ice dominant and water
dominant status usually represent an intermediate stage. Therefore, ambiguities arose when comparing the freeze-up/break-up
results with other lake ice phenology datasets. Moreover, even lake-scale datasets that include four lake ice phenology events
represent different freeze–thaw information due to different lake extensions and thresholds. For example, Cai et al. (2019)
extracted lake ice phenology based on thresholds of 5% and 95%, which might overlook freeze–thaw information near a lake
shore. Wang et al. (2021a) only used one 18.7 GHz AMSR2 pixel closest to the central point of a lake, which means that the
ice phenology extracted only represents the freeze–thaw information from an area of 22 km × 14 km near the lake center.
Furthermore, datasets of lake boundaries used in these studies might be different. Since lakes on the Tibetan Plateau have
undergone rapid changes in recent decades (Yao et al., 2018; Zhang et al., 2021c), this may also result in inconsistent
information contained in lake ice phenology results.

Although there might be inconsistencies with existing lake ice phenology datasets, this new dataset extracted from ASMR2
$T_B$ data has great advantages as an independent dataset. On the one hand, the latest lake boundary was used, and complete ice
phenology records for the largest number of lakes were obtained. Although PMW LIP dataset contains records of 106 lakes
on the Tibetan Plateau, not all the records were complete from 2013 to 2020. Specifically, the number of lakes with complete
eight-year records for freeze onset, complete ice cover, melt onset, and complete ice free, were 81, 28, 11, and 79, respectively.
So, the new dataset at least doubles the record of lake ice phenology on the Tibetan Plateau. On the other hand, pixel-scale
records can provide daily information of the process of ice formation and decay on the lake surface, which is difficult to
achieve for some lakes through fine spatial resolution optical data such as MODIS and AVHRR due to the influence of cloudy
weather. In addition, compared with the two MODIS true-color images under clear sky conditions (one image in the freeze-up



season and another in the break-up season), the spatial distribution of lake ice and open water obtained by AMSR2 data was consistent with the true-color images (Fig. 5). Some differences might be caused by short-term repeated freeze-up and break-

up or the sampling interval of passive microwave data.

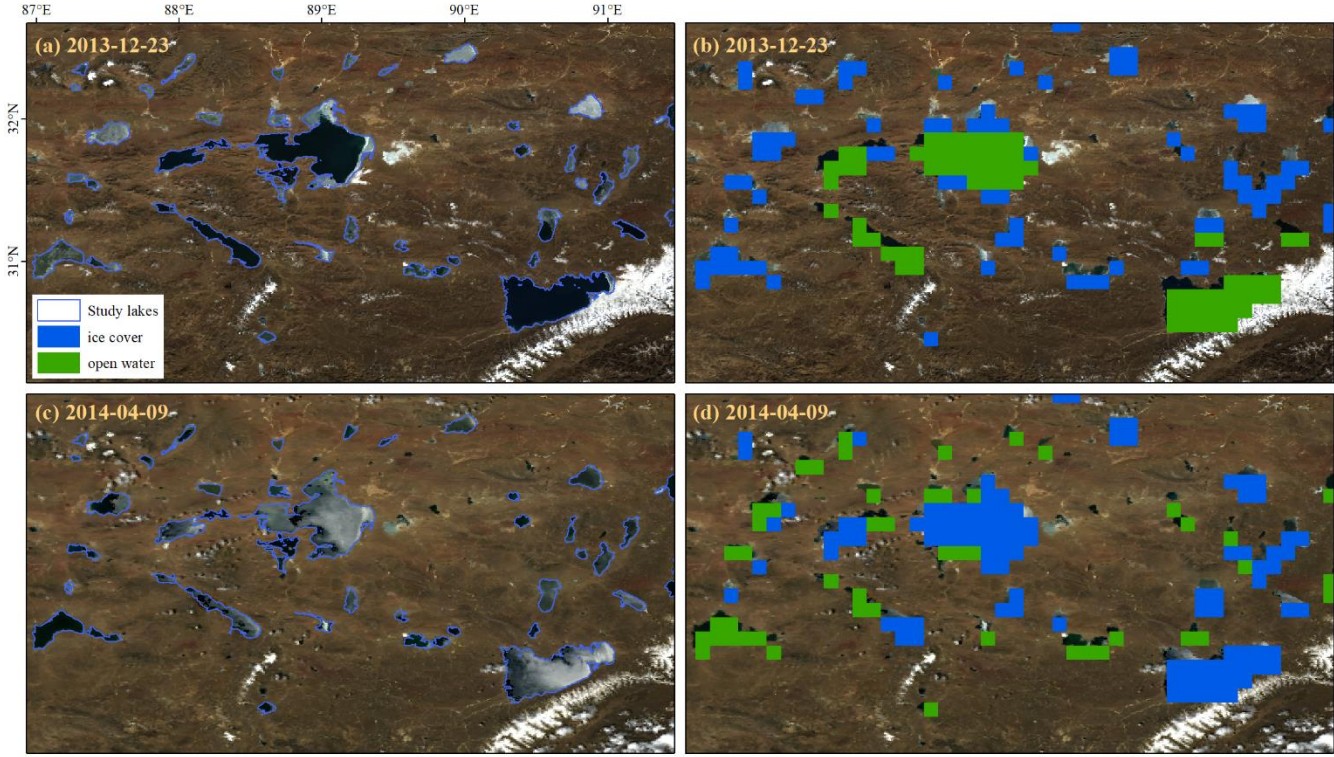

**Figure 5: Comparisons of spatial distribution of AMSR2-derived ice and water pixels and MODIS true-color images (MOD09GA product, Vermote and Wolfe, 2021).**

### 3.1.2 Comparison to MODIS LIP and PMW LIP


Two passive microwave datasets, AMSR2 LIP and PMW LIP, were compared to the optical dataset, MODIS LIP, and the results are shown in Fig. 6. Overall, the FUE, BUS, and BUE of the AMSR2 LIP showed higher levels of consistency with the MODIS LIP than the PMW LIP, especially for break-up dates, which showed better correlations. The FUE obtained from passive microwave data tended to occur earlier than that from MODIS data (Fig. 6b). For the AMSR2 LIP, this might be

because we did not consider the freeze-up process inside a pixel, while for the PMW LIP, this might be because only the central pixel was considered, which might cause large differences for areas with large lakes. Meanwhile, ignoring the break-up process inside a pixel also led to differences in the BUS (later) and BUE (earlier) between the AMSR2 LIP and MODIS LIP. Since break-up usually starts from the center of a lake and ends at the lakeshore, the BUS and BUE of the PMW LIP both occurred earlier than those of the MODIS LIP.





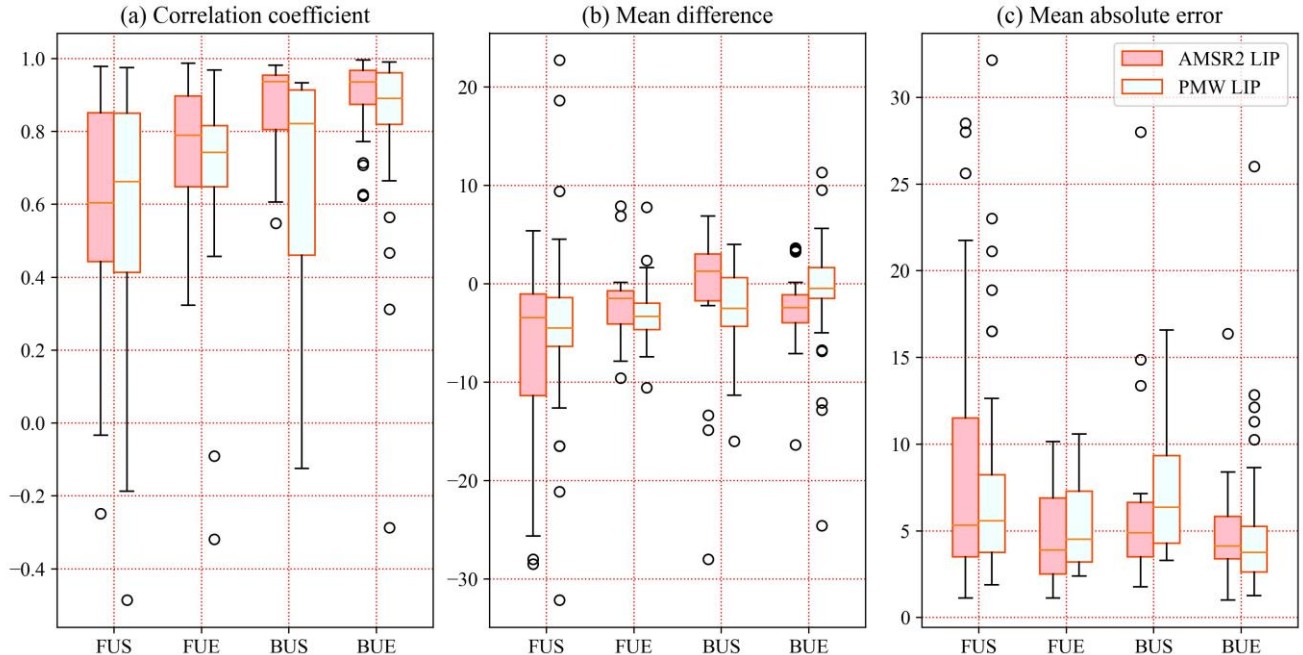

**Figure 6: Comparison between passive-microwave-derived lake ice phenology datasets (AMSR2 LIP and PMW LIP) and the MODIS-derived dataset. (a) Correlation coefficient, (b) mean difference, and (c) mean absolute error.**

However, for FUS, both passive microwave datasets showed less agreement with the MODIS LIP (Fig. 6). FUS obtained from passive microwave data tended to occur earlier than MODIS-derived FUS (Fig. 6b), and several factors might have led to this difference. On the one hand, a threshold of 5% was used to extract FUS from MODIS data, which might overlook early formed lake ice near the lake shore, and sometimes this part of lake ice might last for days or even weeks. On the other hand, we found that some lakes had expanded significantly in recent decades, while newly submerged, shallow beaches were usually located where lake ice first formed. For example, Seling Co has experienced significant expansion in recent years, especially with the lake boundary extending much further to the north, where lake ice formed approximately a month earlier than across the rest of the lake. In addition, some lake groups included lakes with inconsistent freeze–thaw processes, which might affect the lake ice phenology results of the entire lake group.

### 3.1.3 Qinghai Lake group example

We use the Qinghai Lake group, which included Qinghai Lake and its sublake Gahai, as an example to discuss the differences between the lake ice phenology obtained from AMSR2 and MODIS data (Fig. 7). The cloud-gap-filled water pixel series and ice phenology dates of Qinghai Lake and Gahai were taken from Cai et al. (2019). The east and west shores of Qinghai Lake and smaller and shallower lake Gahai were usually the areas to form ice cover earliest. Lake ice here sometimes lasted for weeks; afterward, lake ice started to spread to the entire lake surface of Qinghai Lake, the proportion of lake water pixels was



less than 95% (that is, the lake ice proportion was greater than 5%), and Qinghai Lake would be considered to have started to freeze. We can see that the AMSR2 pixel freezing earliest had a freeze-up date close to the FUS date of Gahai but one much earlier than the FUS date of Qinghai Lake (Fig. 7a). This early formed lake ice, which is usually accompanied by clouds and rain and thus ignored by the threshold method, can be identified earlier by passive microwave pixels that are not subject to weather conditions. However, compared to early forming lake ice, which can last for weeks or even a few months, the spread of lake ice as well as the break-up process are much faster. Since we did not consider the internal freeze-up and break-up

processes of the pixels, the freeze-up dates of the pixels tended to occur earlier than the actual FUE date of the lake. The same effect occurred for break-up dates. The break-up dates of most pixels occurred later than the BUS date of Qinghai Lake and earlier than the BUE date, and only a few pixels captured the initial and final break-up (Fig. 7a). Nevertheless, the freeze-up and break-up dates of these AMSR2 pixels did reflect the freeze–thaw processes at different locations on the lake surface.

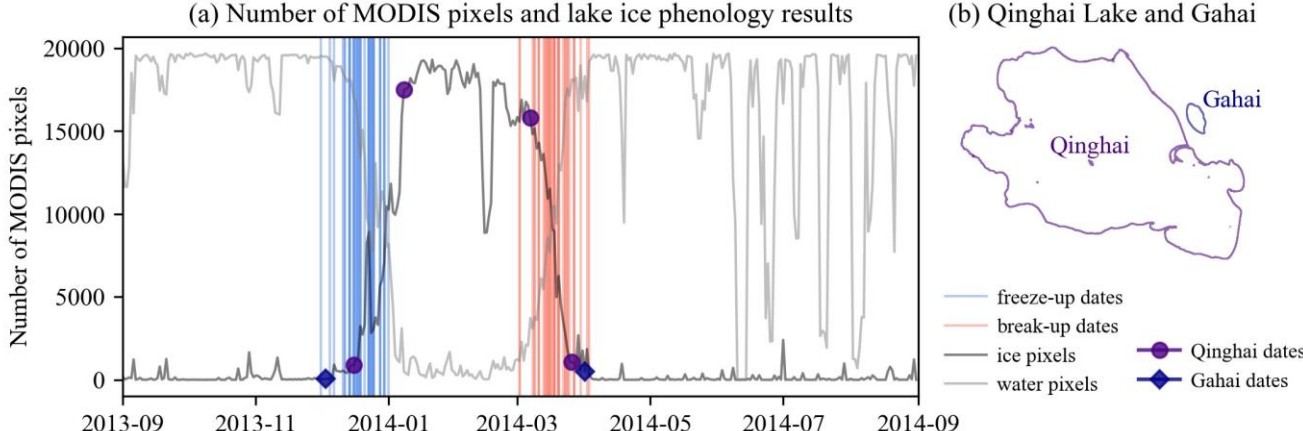

**Figure 7: Comparison of freeze-up and break-up dates derived from multiple AMSR2 pixels in the Qinghai Lake group and the time series of the number of ice/water pixels from the cloud-gap-filled MODIS daily snow product. (a) Number of MODIS ice and water pixels (the number of pixels was summed from the pixels of Qinghai Lake and Gahai), MODIS-derived lake ice phenology for Qinghai Lake (circles, freeze-up start, freeze-up end, break-up start, and break-up end dates) and Gahai (diamonds, freeze-up start and break-up end dates), and freeze-up (blue lines) and break-up (red lines) dates for 51 AMSR2 pixels; (b) shapefile of Qinghai**
**Lake and Gahai.**

### 3.2 Lake ice phenology on the Tibetan Plateau

The dataset provides the freeze-up and break-up dates for each year for September 2012 to August 2022 for each of the 749 pixels, as well as the longitude and latitude and the lake/lake group to which a given pixel belongs. Based on the freeze-up and

break-up dates, we calculated the ice durations for this decade, and the mean values and Sen's slopes were further calculated for each pixel (Fig. 8). Pixels (i.e., lakes) in the northern area usually had longer ice durations than pixels in the southern area (Fig. 8a). Consistent with recent studies (Wang et al., 2019; Cai et al., 2019; 2022b), most southern lakes tended to have shorter





ice durations, while some northern lakes tended to have longer durations (Fig. 8b). However, due to the short temporal coverage, most of the changes were not significant.

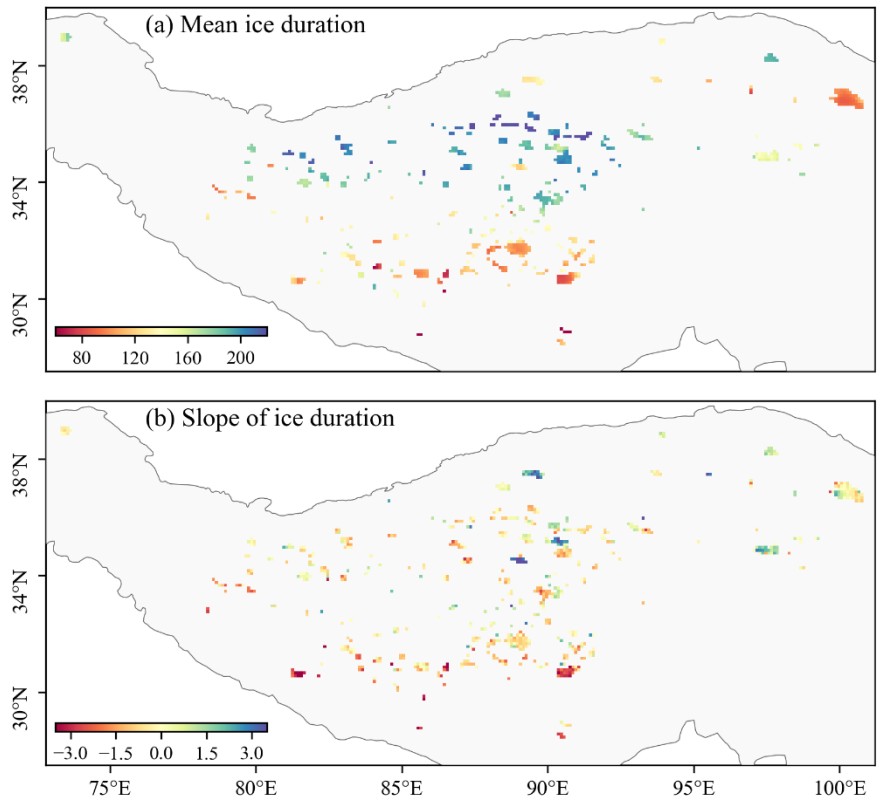


**Figure 8: The mean value (a) and change slope (b) of the ice durations of all pixels.**

## 4 Data availability

The lake ice phenology records for the 749 pixels in 194 lakes are available at https://doi.org/10.11888/Cryos.tpdc.300796 (Cai and Ke, 2023). Calendar dates of freeze-up and break-up dates from September 2012 to August 2022 for each pixel are provided in the dataset. The latitude and longitude of each pixel and the lake/lake group to which each pixel belongs are also recorded in the dataset.

## 5 Conclusions

This study used ASMR2 $T_B$ data and ERA5 air temperature data to extract the ice phenology for 194 lakes on the Tibetan Plateau from September 2012 to August 2022. For each pixel of each lake, we used air temperature to remove seasonal

330 variation in the $T_B$ series and then set thresholds to extract freeze-up and break-up dates. This ice phenology dataset includes the largest number of lakes on the Tibetan Plateau to date.

The main uncertainties of this dataset include the error caused by periodic missing $T_B$ observations and a failure to identify freeze–thaw processes occurring inside pixels. Compared with the existing passive microwave-derived global lake ice phenology dataset, our dataset showed higher consistencies with the MODIS-derived dataset for the Tibetan Plateau region,

335 with the average MAEs of 8.4, 4.5, 6.5, and 4.7 days for FUS, FUE, BUS, and BUE, respectively. The differences between the datasets are mainly related to different definitions of ice phenology events used and the different ranges of lakes represented.

The advantages of this dataset include its provision of ice phenology records for more small lakes, ensuring the integrity of records, and provision of the process of ice formation and decay on the lake surface at the pixel scale. This will help researchers

340 examine lake changes on the Tibetan Plateau, especially with respect to small lakes and lake surface change processes occurring over the past decade.

Finally, AMSR-E is currently not included because there is no altitude-corrected L1R product for AMSR-E data. Regular updates of the lake ice phenology data record are planned with future releases of AMSR2 data and altitude-corrected ASMR-E data, and upcoming AMSR3 data. Work can also be carried out for other passive microwave data, including SMMR, SSM/I-

345 SSMIS, and MWRI data, to extend the temporal coverage of the dataset. In addition, new algorithms are expected be developed to extract lake ice phenology from mixed pixels in the future, such as those based on mathematical models or machine learning and deep learning.

**Author contributions**

YC conceived the idea. YC designed the code and carried out the data processing with contributions from JW, YX, and ZW,

350 YC drafted the manuscript and XS and HL edited it. CQK supervised the study.

**Competing interests**

The authors declare that they have no conflict of interest.

**Acknowledgements**

The authors wish to thank all of the data providers for their data.



**Financial support**

This research has been supported by the National Natural Science Foundation of China (grant no. 42201135) and National Key Research and Development Program of China (grant no. 2022YFC3202104)

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
