# Peer review of "A dataset of 190 lakes' ice phenology on the Tibetan Plateau extracted from AMSR2 data"

_Earth System Science Data, 2023_

## Referee Comment (RC1)

This manuscript introduces a novel dataset detailing the ice phenology of small lakes (< 50km 2) through the integration of Brightness Temperatures (Tb) from passive microwave observations with air temperature (Ta) data sourced from ERA5, specifically focusing on the Tibetan Plateau (TP). The incorporation of air temperature data proves pivotal in mitigating seasonal fluctuations and amplifying the discernible differences of Tb at frozen and ice-free seasons. Overall, the method is interesting but may not be robust for generating a dataset. Please find my questions below.

1. Now it is very common that studies integrate both high-resolution optical data and PMW data to generate high-resolution, continuous snow surface properties, such as snow depth [1], snow albedo phenology [2], snow fraction [3], and snow mask [4,5]. I believe if you follow such similar ideas, you may get better results, and the way Ta and Tbs used in this study is a little bit simplified.

2. The sensitivity test between lake size and the model accuracy should be given. Even though including Ta may remove the seasonal cycle to some degree, it does not mean this proposed method works for all small lakes. Such analysis will give the readers a hint of how robust the model is in different lake sizes.

3. The fundamental assumption underlying this work, as I understand it, is that ERA5 Ta exhibits a closely aligned seasonal cycle with Tb, enabling the removal of this cycle and thereby enhancing the discernible Tb changes. However, in theory, variations in the timings of peaks and valleys in annual cycles of Ta versus surface/soil temperatures are very different. Fig 2b also indicates that the annual valley of Ta is ahead of Tb. How do the authors deal with such a mismatch?

4. Would the post modification (Sect. 2.3.3) be too subjective to affect the proposed data to be used for application (e.g., temporal trend analysis)? A year-to-year temporal variations of the FUS, FUE, BUS, and BUE are needed to test the stability.

5. Reanalysis Ta has a very large uncertainty in TP areas because of the incorrect snow cover simulation [6].

6. Examples in Figure 5 still provide pretty large lakes that are larger than one single pixel thus its phenology won't be very hard to be detected. Some cases for sub-pixel lakes are necessary.

7. In Fig 3b, there are Tbs in ice-free season, especially during 2013.09 – 11, making the first threshold not robust.

8. Any spatial maps of threshold Tbs/dates for different lakes? The map like Fig 8 has few spatial details.

9. Line172: why the thresholds for breakup periods were always higher than those for freeze-up periods

10. The manuscript requires additional accuracy evaluation and data variation analysis, such as the statistics of the lake areas, freeze/thaw date annual variation. The cross validation with MODIS is not enough, any ground measurements? NASA IMS snow/ice cover can be another high-resolution continuous reference data.

**Minor**

The definition of mid- and small lakes should be clarified in the abstract.

Line 101: spell the LIP where it appears for the first time in the article.

Line 94: any processing for the AMSR2 swath gaps? They are very common in TP areas.

Line 97: ERA5-land has much higher spatial resolution compared to ERA5, why not choose ERA5-land?

Line 60, please add one sentence to clarify why previous studies cannot involve mid- and small lakes. As you are using the same input data, why this study can capture the free-thaw info from the subpixel?

As a manuscript for a data journal, this is for end users. Suggest including one or two sentences to introduce why PMW data can capture the freeze-thaw signals.

Include this study in Table 1

Line 79: any specific value range of the object lake areas? how these object lakes were selected? Manually?

[1] Xiong, Chuan, et al. "Mountain Snow Depth Retrieval from Optical and Passive Microwave Remote Sensing Using Machine Learning." IEEE Geoscience and Remote Sensing Letters 19 (2022): 1-5.

[2] Jia, Aolin, et al. "Improved cloudy-sky snow albedo estimates using passive microwave and VIIRS data." ISPRS Journal of Photogrammetry and Remote Sensing 196 (2023): 340-355.

[3] Xiao, Xiongxin, et al. "Estimating fractional snow cover from passive microwave brightness temperature data using MODIS snow cover product over North America." The Cryosphere 15.2 (2021): 835-861.

[4] Kumar, Sujay V., et al. "Quantifying the added value of snow cover area observations in passive microwave snow depth data assimilation." Journal of Hydrometeorology 16.4 (2015): 1736-1741.

[5] Foster, James L., et al. "A blended global snow product using visible, passive microwave and scatterometer satellite data." International journal of remote sensing 32.5 (2011): 1371-1395.

[6] Meng, X., et al. "Simulated cold bias being improved by using MODIS time-varying albedo in the Tibetan Plateau in WRF model." Environmental Research Letters 13.4 (2018): 044028.

---

## Author Comment (AC1)

**Response to RC2**

The study proposed a new method to extract lake ice phenology in passive microwave data by combining ERA-5 air temperature. This is an interesting exploration for passive microwave data, However, as a dataset article, the applicability, advantage and necessity of the dataset still need further consideration and discussion.

**Response:** Thank you for your comments. Combined with the comments from Referee #1, we have re-organized the discussion and validation section in the revised manuscript. First, since our dataset only provides two ice phenology dates (freeze-up and break-up) for each of the pixels, we avoid the FUS/FUE/BUS/BUE terms in the revised manuscript. All the comparisons have been performed on freeze-up and break-up dates. Second, we have added a section of "Advantages and uncertainties of the dataset" to combine the original Uncertainty analysis section, and to discuss the advantages of the dataset and some considerations in the algorithm design. Third, the results of the dataset have been re-evaluated through (1) comparisons with MODIS true-color images, (2) comparisons with daily ice cover series from MODIS/Terra snow cover product and IMS snow and ice product, and (3) comparisons with existing lake ice phenology products. For specific modifications, please see the point-by-point responses to the questions.

Detail questions below:

(1) In the introduction, authors list previous research in Table 1. But as far as I know, there are also some other dataset or studies on lake ice phenology on Tibetan Plateau or even larger area, such as Wu et al. (2022), Feng et al. (2021), and Qiu et al. (2017)…

*Qiu Y., Guo H., et al. A dataset of microwave brightness temperature and freeze-thaw for medium-to-large lakes over the High Asia region (2002-2016). Science Data Bank. 2017. doi: 10.11922/sciencedb.374*

*Wang X., Feng L., et al. High-resolution mapping of ice cover changes in over 33,000 lakes across the North Temperate Zone. Geophysical Research Letters, 2021.*

*Wu Y., Guo L., et al., Ice phenology dataset reconstructed from remote sensing and modelling for lakes over the Tibetan Plateau. Scientific Data, 2022, 9(1):743.*

**Response:** Thank you for providing the three datasets. We have added the results of Wu et al. (2022) and Qiu et al. (2017) to Table 1 and introduce them in the Introduction section in the revised manuscript.

Qiu et al. (2017) obtained the dataset based on AMSR2 data, and the lake ice phenology in Wu et al. (2022) were simulated using models. We also considered using the two datasets for cross-validation, but the overlapping time were a bit short (four years for Qiu et al. (2017) and five years for Wu et al. (2022)), so we gave up.

As for the work of Feng et al. (2021), this work used MODIS data to obtain the maximum annual ice coverage instead of ice phenology dates. We cannot compare the ice cover with ice phenology records because for lakes with annual ice cover, the maximum annual ice coverage would always be 100%. So we didn't add it to Table 1.

(2) Line 73-75 "we drew a 20 km… freeze-thaw information." How did authors determine whether the AMSR2 pixels within the buffer contained extractable freeze–thaw information? By visual check? Or some automatically method?

**Response:** After we used air temperature to reduce the seasonal variations in the $T_B$ series, we visually checked the ten-year $T_B$ series for each pixel to determine whether it contained extractable freeze-thaw information. We have added the statement in the revised manuscript.

(3) The title of section 2.2 is "Input data", but validation data is also included in section 2.2. I suggest as "2.2 Data…2.2.1 Input data…2.2.2 validation data…" and describe AMSR2 and ERA5 data in 2.2.1.

**Response:** Thanks for your suggestion, we have modified it in the revised manuscript.

(4) What is the accuracy of the two validation data? Has the reliability or robustness of the two dataset been validated?

**Response:** For MODIS-derived lake ice phenology, compared with passive microwave datasets derived from AMSR-E/2 and SSM/I, the freeze-up dates had MAEs range from 2.92 to 7.25 days, and the break-up dates had MAEs range from 1.75 to 3.25 days. For PMW-derived lake ice phenology, compared with ground records, complete ice cover and complete ice free dates had correlation coefficients of 0.93 and 0.84, respectively, and the RMSE of 11.84 and 10.07 days, respectively. We have added the statements in the revised manuscript.

(5) As the authors mentioned in the manuscript that optical products (MODIS) and microwave products (PMW) have their own problems and shortcomings, it is not enough to compare the results with these two types of data. Some in-situ data should be collected as much as possible to validate the accuracy of dataset. Or at least, using high-resolution optical data (Landsat, Sentinel…) to visually interpret and obtain reliable lake ice coverage to validate.

**Response:** Unfortunately, we do not have in situ lake ice phenology observations on the Tibetan Plateau. Moreover, we feel that it is difficult to compare the passive microwave-derived results with Landsat and Sentinel data. Because only a few large lakes containing dozens of passive microwave pixels can calculate the lake ice coverage to compare with the optical images.

Instead, we combined the comments from Referee #1, and compared the freeze-thaw dates with daily ice cover series from MODIS/Terra snow cover product and IMS snow and ice

product. The comparisons were performed for three lakes with different sizes (Lake Nam, with an area of 2017.09 km$^2$ and 23 AMSR2 pixels, Lake Gozha, with an area of 249.37 km$^2$ and 5 AMSR2 pixels, and Lake long851lat322, with an area of 51.98 km$^2$ and 1 AMSR2 pixel) (Fig. R1).

[Figure]

**Figure R1:** Comparisons with daily ice cover from MODIS/Terra snow cover product and IMS snow and ice product. Each blue line represents the freeze-up date of a pixel, and green line represents the break-up date. (a), (b), and (c) Freeze-up and break-up dates and daily ice cover of MODIS and IMS data of Lake Nam (23 AMSR2 pixels), Lake Gozha (5 pixels), and Lake long851lat322 (1 pixel), respectively; (d) freeze-up and break-up dates and daily AMSR2 brightness temperature series of Lake long851lat322.

The proportions of lake ice pixels of MODIS and IMS data were normalized by the maximum number of the ice pixels during the ten years. Since Lake Nam and Gozha had multiple pixels covering the lake, the freeze-thaw dates obtained also had multiple records. The results of the AMSR-derived dates had high agreement with the freeze-thaw process shown by the daily ice cover changes of MODIS and IMS data. Since Lake long851lat322 had only one pixel, the daily AMSR2 $T_B$ series were also provided (Fig. R1d). It can be seen that the freezeup and break-up dates were correctly extracted from the highly fluctuated $T_B$ of the mixed pixel, and were consistent with the ice cover changes from MODIS data. However, changes in the lake ice pixels provided by IMS data might be later than the actual freeze-up and break-up events (Fig. R1c). This is because IMS data would not be updated for particular regions when analysts did not have enough information (USNIC, 2008). In addition, it can be seen that due to the influence of cloud cover, the number of lake ice pixels from MODIS data fluctuated constantly, and there were many misclassified ice pixels during the warm seasons, which might bring certain difficulties to the extraction of lake ice phenology. Therefore, for some lakes with persistent cloud cover in cold seasons, it might not be possible to extract lake ice phenology using optical data.

The new figure and the statements have been added to the revised manuscript.

(6) As I understand, the new method eliminates the influence of land in land–water mixed pixels by EAR Ta. Is the method adaptable to all land–water mixed pixels? Even for those pixels that are primarily contained by land.

**Response:** The method was performed to all pixels to remove the seasonal variations in the $T_B$ series to enhance the $T_B$ difference between the ice-covered season and ice-free season before extracting lake ice phenology.

While in the extraction, the automatic threshold calculated from the extreme $T_B$ values were not suitable for all pixels, because the $T_B$ values of the pixels could vary greatly under different ground surface conditions. Therefore, we checked ten years of freeze-thaw results for each pixel and manually corrected the dates for which the automated thresholds were not successfully extracted. Nevertheless, both automatic threshold extraction and manual extraction were performed based on the Gaussian-filtered $\Delta T_B$ series, which was adjusted by the air temperature series to remove the seasonal variation and enhance the $T_B$ difference between the ice-covered season and ice-free season. During manual correction, the results extracted from automated thresholds would be used as a reference to obtain more comparable freeze-up and break-up dates.

We have added the statements in the discussion section in the revised manuscript.

(7) I'm confused in how the threshold are determined. There are 4 phenology indicators (FUS, FUE, BUS, BUE) need to be extracted, which should correspond to 4 THs. In Line 161-162 "we obtained … averaging the two mean values". The average value of two groups mean value is the threshold. Which indicator is this threshold for? A clearer description is needed here.

**Response:** Our dataset only provides two phenology indicators (freeze-up and break-up), so only this threshold was used. The four phenology indicators were only calculated to make the comparison with existing lake ice phenology products.

Therefore, in order to avoid confusion, in the revised manuscript, we no longer calculate the four dates of FUS, FUE, BUS, and BUE, but only use freeze-up and break-up dates. Taking Fig. R1 as an example, in the comparison of three example lakes, we give the freeze-up and break-up dates of all pixels.

(8) In section 2.3.4, Is the lake group comparable to a single lake? Because even lakes that are very close in distance may have different freeze-thaw properties. Perhaps comparing the LIP of the same single lake is more convincing.

**Response:** Thank you very much for your suggestion. We also think that the lake groups should be excluded. In addition, we checked the lake boundaries used in our dataset, MODIS LIP, and PMW LIP, and lakes with large boundary changes were also excluded.

Among the 71 lakes contained in the MODIS LIP, our dataset contained 64 lakes, 27 of which were matched to lake groups, and 9 lakes had boundaries with large differences, so the remaining 28 lakes were compared. As for the PMW LIP, our dataset contained 105 of 106 lakes in the product, 50 of which were matched to lake groups, and 13 lakes had boundaries with large differences, so the remaining 42 lakes were compared.

Since the number of lakes included in both validation datasets was relatively small after the screening, the comparison was no longer performed on three datasets together. Instead, we compared our dataset with MODIS LIP and PMW LIP separately.

In the comparison, if the validation product had both beginning and end records of freeze-up, we calculated the mean dates to compare with the median freeze-up dates of all pixels of the lake (same for break-up dates). If the validation product only provided the beginning dates of freeze-up, we calculated the earliest freeze-up dates of all pixels of the lake to compare; if only the end dates of break-up were provided, the latest break-up dates of all pixels were calculated to compare. For each lake, the correlation coefficient, mean difference (MODIS LIP or PMW LIP minus AMSR2-derived results), and mean absolute error were calculated (Fig. R2).

[Figure]

**Figure R2:** Comparison with MODIS-derived and PMW-derived lake ice phenology datasets

(MODIS LIP and PMW LIP). (a), (b), and (c) Correlation coefficient (r), mean difference (bias, MODIS LIP minus AMSR2-derived results), and mean absolute error (MAE) of all lakes compared with MODIS LIP, (d), (e), and (f) r, bias (PMW LIP minus AMSR2-derived results), MAE of all lakes compared with PMW LIP.

Except for the boxplots, we also compared the year-to-year temporal variations of freeze-up and break-up dates for the three lakes mentioned in Fig. R1 (Lake Nam, Gozha, and long851lat322) (Fig. R3).

[Figure]

**Figure R3:** Comparisons of annual freeze-thaw records with MODIS-derived and PMW-derived lake ice phenology datasets. (a) and (b) Comparison of the freeze-up and break-up dates of Lake Nam, (c) and (d) comparisons of Lake Gozha, and (e) and (f) comparisons of Lake long851lat322. MODIS dates and PMW dates with one line represent the beginning of freeze-up or the end of break-up, while two lines in (c) represent the beginning and end of freeze-up, respectively. The y-axis means the day of year of 1 September.

Overall, the lake ice phenology time series from the three datasets had relatively high consistency, especially compared the AMSR2-derived results to the MODIS LIP. For large lakes, pixel-scale freeze-up and break-up records can give more detailed information than traditional lake-scale records. Taking Lake Nam as an example, there are some pixels had earlier freeze-up dates than the records from MODIS LIP. In the extraction algorithm of MODIS LIP, a 5% threshold was used to extract the beginning dates of freeze-up to avoid the impact of repeated freeze-thaw events, which might overlook early freeze-up information near a lake shore. In contrast, break-up process usually involves less repeated freeze-thaw, so the latest

break-up dates derived from AMSR2 data had good consistent with the end dates of break-up in the MODIS LIP. However, the PMW LIP only used one pixel closest to the central point of a lake, and sometimes it could not obtain the freeze-thaw information of the entire lake, especially for lakes with large areas. While for lakes with smaller areas, since the PMW LIP recorded the beginning date of freeze-up and end dates of break-up, it might obtain earlier freeze-up dates and later break-up dates than this dataset (Fig. R3e-f). In addition, the records of the PMW LIP were incomplete for some lakes. For example, the PMW LIP only had six records for the break-up dates of Lake Gozha (Fig. R3d). Therefore, although the same AMSR2 data were used, this new dataset could provide complete records for more lakes than the PMW LIP, and had good consistency with the MODIS LIP records.

The new figures and the corresponding statements have been added to the revised manuscript. And since Fig. R1 and Fig. R3 have already shown the comparisons of three single lakes, we have deleted the section of Qinghai Lake group example.

(9) Line 224-225, so how to determine the freeze-thaw information in pixels with high proportion of land? By visual interpretation?

**Response:** Here is an explanation of why we did not extract four ice phenology dates in our datasets (because it is difficult to extract more detailed beginning and end information of freeze-thaw from mixed pixels). To avoid confusion, we have removed this statement in the revised manuscript.

(10) I'm confused how much lake ice indicator are extracted. Two or four? Please state in Section 2.3. If only two indices are considered, how to compare them with dataset that contained 4 indicators (Such as Figure 6)?

**Response:** We only extracted 2 dates. In the original comparison, we obtained the maximum and minimum values in all freeze-up and break-up dates for each lake to make the comparisons (L208-211). However, in order to avoid confusion, we have avoided the description of the four dates in the revised manuscript and redone the comparisons (please see the response to Comment 8).

(11) In conclusion, line 338-340. The authors mentioned the dataset contained more small lakes. How small lakes that the LIP can be extracted? Because the resolution of passive microwave data is coarse, is the dataset more accurate than that based on optical data in small lakes?

**Response:** Combined with the comments from Referee #1, we calculated the distribution of area and number of pixels of the study lakes (Fig. R4).

[Figure]

**Figure R4.** The distribution of area and number of pixels of the study lakes. (a) The distribution of area of 194 study lakes, (b) the distribution of area of 153 lakes or lake groups. (c) the boxplots of number of pixels for lakes in different area intervals and the average proportion of pixels automatically extracted by the threshold.

Among the 194 study lakes, there were 39 small lakes (area $< 50$ km$^2$), of which the smallest one had an area of 14.66 km$^2$, 137 medium-sized lakes (50–500 km$^2$), and 18 large lakes ($> 500$ km$^2$). The largest lake was Lake Qinghai, with an area of 4541.43 km$^2$ (Fig. R4a). After grouping, the lakes/lake groups were still mainly small and medium-sized. The smallest single lake had an area of 21.92 km$^2$ (Fig. R4b).

In addition, from the comparisons with MODIS daily ice cover series (Fig. R1), we can see that MODIS data are sometimes severely affected by cloud cover during the cold seasons, making it difficult to determine lake ice phenology. For example, MODIS-derived product did not contain the records for Lake Zonag because this lake had an average cloud cover of 73% in January and 71% in February from 2013 to 2022. Passive microwave data are not affected by weather conditions. Therefore, we obtained the lake ice phenology of 130 more lakes than the MODIS-derived lake ice phenology product. We believe that passive microwave can obtain more accurate results when affected by cloudy weather. In contrast, MODIS can sometimes obtain the lake ice phenology of smaller lakes. There are still 7 lakes in the MODIS-derived product that are not included in our dataset, and three of them have an area of only 10-11 km$^2$.

(12) The applicability, and applicable scenarios of the dataset need to be further clarified in the conclusion

**Response:** Thanks for your suggestion. The dataset is available to users to investigate the spatial distribution, change trends and influencing factors of lake ice phenology on the Tibetan Plateau under the background of global climate change. We have added the statement in the revised manuscript.

---

## Author Comment (AC2)

**Response to RC1**

This manuscript introduces a novel dataset detailing the ice phenology of small lakes (< 50km 2) through the integration of Brightness Temperatures (Tb) from passive microwave observations with air temperature (Ta) data sourced from ERA5, specifically focusing on the Tibetan Plateau (TP). The incorporation of air temperature data proves pivotal in mitigating seasonal fluctuations and amplifying the discernible differences of Tb at frozen and ice-free seasons. Overall, the method is interesting but may not be robust for generating a dataset. Please find my questions below.

**Response:** Thank you for your comments. In this study, we try to find an easy and efficient way to extract lake ice phenology for more lakes using passive microwave data. We believe that although the method is simple, it basically enables us to achieve the purpose of this study. Combined with the comments from Referee #2, we have re-organized the discussion and validation section in the revised manuscript. We have added a section of "Advantages and uncertainties of the dataset" to combine the original Uncertainty analysis section, and to discuss the advantages of the dataset and some considerations in the algorithm design. The results of the dataset have been re-evaluated through (1) comparisons with MODIS true-color images, (2) comparisons with daily ice cover series from MODIS/Terra snow cover product and IMS snow and ice product, and (3) comparisons with existing lake ice phenology products. For specific modifications, please see the point-by-point responses to the questions.

1. Now it is very common that studies integrate both high-resolution optical data and PMW data to generate high-resolution, continuous snow surface properties, such as snow depth [1], snow albedo phenology [2], snow fraction [3], and snow mask [4,5]. I believe if you follow such similar ideas, you may get better results, and the way Ta and Tbs used in this study is a little bit simplified.

**Response:** Thanks for providing these studies. We've read these papers and learned that there are great advantages in integrating different types of remote sensing data and using novel methods such as machine learning. In future work, we will further learn from these studies and hope to obtain better lake ice phenology datasets.

For this paper, we realized that our method is relatively simple, but we believe that this dataset can provide better results for more lakes than other existing lake ice phenology datasets. We have carefully modified the manuscript, especially rearranging the validation and discussion sections, hoping to make the results of this dataset more convincing.

2. The sensitivity test between lake size and the model accuracy should be given. Even though including Ta may remove the seasonal cycle to some degree, it does not mean this proposed method works for all small lakes. Such analysis will give the readers a hint of how robust the model is in different lake sizes.

**Response:** Before extracting lake ice phenology, the same algorithm was performed to 100% of the pixels to remove the seasonal variations in the $T_B$ series to enhance the $T_B$ difference between the ice-covered season and ice-free season. While in the extraction, the automatic threshold calculated from the extreme $T_B$ values might not be suitable for all pixels, because the $T_B$ values of the pixels could vary greatly under different ground surface conditions. Therefore, we checked ten years of freeze-thaw results for each pixel and manually corrected the dates for which the automated thresholds were not successfully extracted. Nevertheless, both automatic threshold extraction and manual extraction were performed based on the Gaussian-filtered $\Delta T_B$ series, which was adjusted by the air temperature series to remove the seasonal variation and enhance the $T_B$ difference between the ice-covered season and ice-free season. During manual correction, the results extracted from automated thresholds would be used as a reference to obtain more comparable freeze-up and break-up dates.

We calculated the distribution of area and number of pixels of the study lakes to help the discussion, and the results are shown in Fig. R1.

[Figure]

**Figure R1.** The distribution of area and number of pixels of the study lakes. (a) The distribution of area of 194 study lakes, (b) the distribution of area of 153 lakes or lake groups. (c) the boxplots of number of pixels for lakes in different area intervals and the average proportion of pixels automatically extracted by the threshold.

This figure highlights that our study lakes are mainly small and medium-sized lakes, and on the other hand, we calculated the proportion of pixels automatically extracted by the threshold. In general, the smaller the lake area (the more mixed pixels in the lake/lake group), the lower the automatic extraction ratio would be (Fig. R1c). For all freeze-up and break-up records from 2013 to 2023 (14,942 records), the proportion of automatic extraction was 84.96%, with a relative lower proportion of 80.77% for freeze-up dates and 87.16% for break-up dates.

This is because the freeze-up process of lakes usually takes longer than the break-up process and is more prone to short-term repeated freeze-up and break-up, making the whole process more complicated, and thus more difficult to determine automatically.

The new figure and corresponding statements have been added to the revised manuscript.

3. The fundamental assumption underlying this work, as I understand it, is that ERA5 Ta exhibits a closely aligned seasonal cycle with Tb, enabling the removal of this cycle and thereby enhancing the discernible Tb changes. However, in theory, variations in the timings of peaks and valleys in annual cycles of Ta versus surface/soil temperatures are very different. Fig 2b also indicates that the annual valley of Ta is ahead of Tb. How do the authors deal with such a mismatch?

**Response:** Since the $T_B$ of mixed pixels have much larger fluctuations than air temperature data, slight mismatches will be ignored. Especially after using the cubic polynomial fitting to fit the air temperature curve, the impact of these mismatches will become much smaller.

There have been studies using pure land pixels to decompose the water component in land-water mixed pixels. However, in the lake-rich regions on the Tibetan Plateau, it is difficult to find pure land pixels near lakes. Moreover, the mixed pixel decomposition algorithm will be more complex. In contrast, ERA5-land air temperature data are easily accessible, and the algorithm can achieve the purpose of removing seasonal $T_B$ variations simply and efficiently.

We have added the statements in the revised manuscript to address the uncertainties of ERA5-land air temperature data and explain why we use air temperature in the study.

4. Would the post modification (Sect. 2.3.3) be too subjective to affect the proposed data to be used for application (e.g., temporal trend analysis)? A year-to-year temporal variations of the FUS, FUE, BUS, and BUE are needed to test the stability.

**Response:** As we responded to Comment 2, we checked ten years of freeze-thaw results for each pixel and manually corrected the dates for which the automated thresholds were not successfully extracted. During manual correction, the results extracted from automated thresholds would be used as a reference to obtain more comparable freeze-up and break-up dates.

Combined with the comments from Referee #2, we have redone the comparison with existing lake ice phenology products. To avoid the confusion about lake ice phenology terms, all the comparisons are based on two dates (freeze-up and break-up). And to avoid unreasonable comparisons caused by the changing lake boundary, we checked the lake boundary data used by our dataset, MODIS LIP, and PMW LIP, and screened lakes with consistent lake boundaries for comparison. The new boxplots for all lakes are shown in Fig. R2.

[Figure]

**Figure R2:** Comparison with MODIS-derived and PMW-derived lake ice phenology datasets (MODIS LIP and PMW LIP). (a), (b), and (c) Correlation coefficient (r), mean difference (bias, MODIS LIP minus AMSR2-derived results), and mean absolute error (MAE) of all lakes compared with MODIS LIP, (d), (e), and (f) r, bias (PMW LIP minus AMSR2-derived results), MAE of all lakes compared with PMW LIP.

Except for the boxplots, we compared the year-to-year temporal variations of freeze-up and break-up dates for three lakes with different sizes (Lake Nam, with an area of 2017.09 km$^2$ and 23 AMSR2 pixels, Lake Gozha, with an area of 249.37 km$^2$ and 5 AMSR2 pixels, and Lake long851lat322, with an area of 51.98 km$^2$ and 1 AMSR2 pixel) (Fig. R3).

[Figure]

**Figure R3:** Comparisons of annual freeze-thaw records with MODIS-derived and PMW-derived lake ice phenology datasets. (a) and (b) Comparison of the freeze-up and break-up dates of Lake Nam, (c) and (d) comparisons of Lake Gozha, and (e) and (f) comparisons of Lake long851lat322. MODIS dates and PMW dates with one line represent the beginning of freezeup or the end of break-up, while two lines in (c) represent the beginning and end of freeze-up, respectively. The y-axis means the day of year of 1 September.

Overall, the lake ice phenology time series from the three datasets had relatively high consistency, especially compared the AMSR2-derived results to the MODIS LIP. For large lakes, pixel-scale freeze-up and break-up records can give more detailed information than traditional lake-scale records. Taking Lake Nam as an example, there are some pixels had earlier freeze-up dates than the records from MODIS LIP. In the extraction algorithm of MODIS LIP, a 5% threshold was used to extract the beginning dates of freeze-up to avoid the impact of repeated freeze-thaw events, which might overlook early freeze-up information near a lake shore. In contrast, break-up process usually involves less repeated freeze-thaw, so the latest break-up dates derived from AMSR2 data had good consistent with the end dates of break-up in the MODIS LIP. However, the PMW LIP only used one pixel closest to the central point of a lake, and sometimes it could not obtain the freeze-thaw information of the entire lake, especially for lakes with large areas. While for lakes with smaller areas, since the PMW LIP recorded the beginning date of freeze-up and end dates of break-up, it might obtain earlier freeze-up dates and later break-up dates than this dataset (Fig. R3e-f). In addition, the records of the PMW LIP were incomplete for some lakes. For example, the PMW LIP only had six records for the break-up dates of Lake Gozha (Fig. R3d). Therefore, although the same AMSR2 data were used, this new dataset could provide complete records for more lakes than the PMW LIP, and had good consistency with the MODIS LIP records.

The new figures and the statements have been added to the revised manuscript.

5. Reanalysis Ta has a very large uncertainty in TP areas because of the incorrect snow cover simulation [6].

**Response:** In fact, we only need the air temperature to provide approximate seasonal variation information, and do not require it to be very precise. As we responded to Comment 3, compared to the uncertainties of air temperature data, the $T_B$ of mixed pixels have much larger fluctuations.

We know that the accuracy of reanalysis data on the Tibetan Plateau has always been questionable. In our previous work, we did a simple evaluation of ERA5 data. We collected data from seven meteorological stations to assess the performance of ERA5 data in the region of lakes. The result is shown in the first column of Fig. R4. For air temperature data, all seven stations showed significant consistency between ERA5 data and in situ observations, with the MAE ranging from 0.58 to 1.71 °C. Although there will be inevitable differences between site observations and grid estimates, we believe the consistencies could support variation analysis in this study.

[Figure]

**Figure R4.** Comparisons of air temperature, wind speed and precipitation between ERA5 and station records. (The figure is from Cai et al., 2022)

*Cai, Y., Ke, C.-Q., Xiao, Y., and Wu, J.: What caused the spatial heterogeneity of lake ice phenology changes on the Tibetan Plateau?, Sci. Total Environ., 836, 155517, https://doi.org/10.1016/j.scitotenv.2022.155517, 2022.*

6. Examples in Figure 5 still provide pretty large lakes that are larger than one single pixel thus its phenology won't be very hard to be detected. Some cases for sub-pixel lakes are necessary.

**Response:** For better comparison, we have re-organized the validation section. We gave examples of three lakes with different sizes (as mentioned in Comment 4). Except for the comparisons with existing lake ice phenology products, for the three lakes, we also compared the freeze-up and break-up dates with daily ice cover from MODIS/Terra snow cover product and IMS snow and ice product. Please see the reply to Comment 10. As for Figure 5, We have retained and modified it to help explain how the lakes are divided into lake groups.

[Figure]

**Figure R5** (Figure 5 in the original manuscript): Comparisons of spatial distribution of AMSR2-derived ice and water pixels and MODIS true-color images (MOD09GA product, Vermote and Wolfe, 2021). Each different colored outline represents a single lake or a lake group.

7. In Fig 3b, there are Tbs in ice-free season, especially during 2013.09 – 11, making the first threshold not robust.

**Response:** Because the $\Delta T_B$ from September to November were still fluctuating around 0, we do not think it has frozen during this period. However, sometimes lake does have a small area of repeated freeze-up and break-up before the extracted freeze-up date. This is a characteristic of lake ice which can be observed in all types of remote sensing data. We can only ensure that the date extraction method is consistent to ensure that the results are comparable.

8. Any spatial maps of threshold Tbs/dates for different lakes? The map like Fig 8 has few spatial details.

**Response:** Our dataset provides pixel-scale freeze-thaw dates, we did not calculate the dates for each lake (except for the comparison with other lake ice phenology products). We think pixel-scale dataset is more flexible, and if needed, users can match the pixels to their own lake boundaries based on the geographical coordinates of the pixels. The spatial positions of the 749 pixels are as shown in Figure 8.

For freeze-up and break-up dates, we have added a new figure (Fig. R6) of the results in 2014 in Section 2.3.3 as an example.

[Figure]

**Figure R6:** The freeze-up (a) and break-up (b) dates for 749 pixels in 2014. The dates are calculated as the day of year of 1 September.

9. Line172: why the thresholds for breakup periods were always higher than those for freeze-up periods

**Response:** We explained the reasons in Lines 163-168: "As ice formation requires colder temperatures than decay, temperatures (both air temperature and $T_B$) during break-up periods tend to be higher than those during freeze-up periods. In addition, as ice thickness increases in the winter, the $T_B$ will increase further. As a result, even if seasonal variation were reduced for

the $\Delta T_B$ series, the overall series might still be tilted, especially for pixels with longer ice periods (Fig. 3). Therefore, the threshold used to determine ice status during the break-up periods should be slightly higher than the threshold used for freeze-up periods."

10. The manuscript requires additional accuracy evaluation and data variation analysis, such as the statistics of the lake areas, freeze/thaw date annual variation. The cross validation with MODIS is not enough, any ground measurements? NASA IMS snow/ice cover can be another high-resolution continuous reference data.

**Response:** Thanks for the suggestion. We have re-organized the validation. Except for the modification of the comparisons with existing lake ice phenology, we also compared the freeze-up and break-up dates with the daily ice cover from MODIS/Terra snow cover product and IMS snow and ice product. The comparisons were performed for three lakes with different sizes (Lake Nam, Gozha, and long851lat322, same as the lakes mentioned in Comment 4) (Fig. R7).

[Figure]

**Figure R7:** Comparisons with daily ice cover from MODIS/Terra snow cover product and IMS snow and ice product. Each blue line represents the freeze-up date of a pixel, and green line represents the break-up date. (a), (b), and (c) Freeze-up and break-up dates and daily ice cover of MODIS and IMS data of Lake Nam (23 AMSR2 pixels), Lake Gozha (5 pixels), and Lake

long851lat322 (1 pixel), respectively; (d) freeze-up and break-up dates and daily AMSR2 brightness temperature series of Lake long851lat322.

The proportions of lake ice pixels of MODIS and IMS data were normalized by the maximum number of the ice pixels during the ten years. Since Lake Nam and Gozha had multiple pixels covering the lake, the freeze-thaw dates obtained also had multiple records. The results of the AMSR-derived dates had high agreement with the freeze-thaw process shown by the daily ice cover changes of MODIS and IMS data. Since Lake long851lat322 had only one pixel, the daily AMSR2 $T_B$ series were also provided (Fig. R7d). It can be seen that the freeze-up and break-up dates were correctly extracted from the highly fluctuated $T_B$ of the mixed pixel, and were consistent with the ice cover changes from MODIS data. However, changes in the lake ice pixels provided by IMS data might be later than the actual freeze-up and break-up events (Fig. R7c). This is because IMS data would not be updated for particular regions when analysts did not have enough information (USNIC, 2008). In addition, it can be seen that due to the influence of cloud cover, the number of lake ice pixels from MODIS data fluctuated constantly, and there were many misclassified ice pixels during the warm seasons, which might bring certain difficulties to the extraction of lake ice phenology. Therefore, for some lakes with persistent cloud cover in cold seasons, it might not be possible to extract lake ice phenology using optical data.

The new figure and the statements have been added to the revised manuscript.

Minor

The definition of mid- and small lakes should be clarified in the abstract.

**Response:** We have added the definition in the abstract. In addition, we have changed the "small lakes" in L20 and 22 to "small and medium-sized lakes" to make the statement more accurate.

Line 101: spell the LIP where it appears for the first time in the article.

**Response:** We have added it in the revised manuscript.

Line 94: any processing for the AMSR2 swath gaps? They are very common in TP areas.

**Response:** We performed linear interpolation during the extraction process for the gaps (L157-158). Then in post-processing, if the original $T_B$ of the extracted freeze-up date was missing, the first date after that day with a valid value was recorded. For the break-up date, the latest date before that day with a valid value was recorded if the original $T_B$ was missing (L181 & 183). In the Tibetan Plateau region, even for lakes at the lowest latitude, the sampling interval

of the AMSR2 data generally only spanned one day. So, the error caused by periodic missing data would not exceed one day. We have mentioned the uncertainties caused by the gaps in the uncertainty analysis (L227-228).

Line 97: ERA5-land has much higher spatial resolution compared to ERA5, why not choose ERA5-land?

**Response:** We used ERA5-Land data with a resolution of 0.1°. Sorry for not making it clear, we have modified it in the revised manuscript.

Line 60, please add one sentence to clarify why previous studies cannot involve mid- and small lakes. As you are using the same input data, why this study can capture the free-thaw info from the subpixel?

**Response:** Previous studies usually just excluded land-contaminated pixels and used only pure lake pixels before algorithm design. Therefore, the application of passive microwave data on small and medium-sized lakes was limited. We have added the statements in the Introduction section.

As a manuscript for a data journal, this is for end users. Suggest including one or two sentences to introduce why PMW data can capture the freeze-thaw signals.

**Response:** Thanks for your suggestion, we have added the statements in the Introduction section: "The emissivity of ice is much higher than that of water, so when the lake is covered with ice, the brightness temperature will increase significantly. Based on such differences, passive microwave data can be used to extract lake freeze-thaw information."

Include this study in Table 1

**Response:** We have added it in Table 1 in the revised manuscript.

Line 79: any specific value range of the object lake areas? how these object lakes were selected? Manually?

**Response:** We described the selection process in L73-74: "For each lake larger than 10 km$^2$, we drew a 20 km buffer outward from the boundary and determined whether the AMSR2 pixels within the buffer contained extractable freeze–thaw information." We visually checked the ten-year $T_B$ series for each pixel to determine whether it contained extractable freeze-thaw information. We have clarified the statement in the revised manuscript.